# The adaptive landscape of a metallo-enzyme is shaped by environment-dependent epistasis

Dave W. Anderson[1,2 ✉], Florian Baier[1], Gloria Yang [1] & Nobuhiko Tokuriki [1 ✉]

Enzymes can evolve new catalytic activity when environmental changes present them with novel substrates. Despite this seemingly straightforward relationship, factors other than the direct catalytic target can also impact adaptation. Here, we characterize the catalytic activity of a recently evolved bacterial methyl-parathion hydrolase for all possible combinations of the five functionally relevant mutations under eight different laboratory conditions (in which an alternative divalent metal is supplemented). The resultant adaptive landscapes across this historical evolutionary transition vary in terms of both the number of "fitness peaks" as well as the genotype(s) at which they are found as a result of genotype-by-environment interactions and environment-dependent epistasis. This suggests that adaptive landscapes may be fluid and molecular adaptation is highly contingent not only on obvious factors (such as catalytic targets), but also on less obvious secondary environmental factors that can direct it towards distinct outcomes.

---

[1] Michael Smith Laboratories, University of British Columbia, Vancouver, BC, Canada. [2] Alberta Children's Hospital Research Institute, Department of Molecular Biology and Biochemistry, Cumming School of Medicine, University of Calgary, Calgary, AB, Canada. ✉email: david.anderson1@ucalgary.ca; tokuriki@msl.ubc.ca

Enzyme evolution is fundamentally dynamic, encompassing the myriad of ways in which enzymatic functions change from one state into another[1–3]. Enzymes undergoing the classic Darwinian model of adaptation can be visualized as mutational pathways across adaptive landscapes from the initial genotype to the fittest genotype (i.e., that which exhibits the optimal enzyme activity for its target substrate) through the acquisition of function-altering mutations[4,5]. Epistasis, which arises from interactions between genetic mutations ($G \times G$), can result in pathways that are difficult to predict and fairly restricted, as the effect of each mutation may depend on the presence or absence of other mutations. Many previous studies characterized the adaptive landscapes of proteins and enzymes by functionally assaying all possible combinations of mutations that are responsible for the adaptation of enzymatic function[6–11]. These studies make it clear that epistasis is highly prevalent, which suggests a common ruggedness for many adaptive landscapes. Thus, evolution can be expected to proceed through relatively restricted mutational pathways; some pathways may reach the global "fitness peak" (i.e., the optimal genotype across the adaptive landscape) while others may become stuck on a local peak (a genotype that is suboptimal compared to the global peak)[12–14]. Such ruggedness of the fitness landscape also has important implications for evolutionary phenomena such as repeatability[15,16], contingency[17–20], and (ir)reversibility[21–23]. But how fixed are these adaptive landscapes? Can they be reshaped or altered by variations in nonselective or secondary environmental factors, such as temperature, salinity, pH, the presence of other proteins, or cofactor availability, such as metals? These factors do not necessarily define the novel adaptive function, but they can nonetheless impact the fitness of a genotype (i.e., genotype-by-environment ($G \times E$) interactions)[24,25], and epistasis between mutations (i.e., environment-dependent epistasis ($G \times G \times E$) interactions)[26], and thus, the topology of the adaptive landscapes[27,28]. While several enzyme studies have addressed the impact of "primary" environments (i.e., different substrates or ligands) on the topology of the adaptive landscapes[29,30], the degree to which the nonselective environmental factors can alter evolutionary outcomes even under the same primary selective pressure remains poorly understood[24,31].

We explore these questions and concepts in detail by characterizing the evolutionary transition between an ancestral dihydrocoumarin hydrolase (DHCH) and its methyl-parathion hydrolase (MPH) descendant within the metallo-β-lactamase superfamily[32,33]. This enzyme adaptation occurred in bacteria between the 1940s and the 2000s, coinciding with the human application of organophosphate pesticides in industry and agriculture, thus providing an excellent case of classic Darwinian adaptation[34,35]. MPH was first identified from soil bacteria, *Pseudomonas sp.* WBC-3 that were isolated from soil contaminated with methyl-parathion. Our previous work characterized a set of five mutations—four single-amino acid substitutions and one single-residue insertion that surround the active site (*l*72R, Δ193S, *h*258L, *i*271T, and *f*273L; Fig. 1a)—that is both necessary and sufficient to recapitulate the evolution of the derived MPH activity[11]. MPH requires two divalent ions to be coordinated in the active site in order to be catalytically active (Fig. 1b)[11,36,37]. Whereas the majority of research on MPH to date assume it solely or primarily functions using $Zn^{2+}$ to coordinate the substrate in its active site[38], MPH has also been shown to exhibit varying enzymatic activity and promiscuity when other divalent metals are present[36,37].

Here, we investigate the impact of variation in a secondary environmental factor, specifically the type of metal ion present[33], on adaptation from the DHCH ancestor. We systematically characterize the same genotypes (a complete combinatorial set of those five historical mutations—32 genotypes in total) in order to further examine and compare the adaptive landscapes for each metal environment (Supplementary Fig. 1). By applying extensive statistical analyses, we effectively describe the extent to which variation in metal ion has an impact on the functional effect of individual mutations and the epistasis between them. We subsequently show how metal variation can alter the evolutionary trajectories and outcomes across this particularly relevant adaptive landscape for MPH. This has general implications for the effect of nonselective, secondary environments on protein evolution.

## Results

**Different metal environments alter the evolution of MPH activity.** As MPH has evolved toward degrading methyl-parathion, the presence of the substrate can be considered as the primary selection pressure for the enzyme's evolution. In this study[11,36–38], we define the secondary environment as the abundance of metal ions in the environment because metal ions can affect the activity level of MPH but does not impose a direct selection pressure. We selected eight different divalent metals (calcium—$Ca^{2+}$, cadmium—$Cd^{2+}$, cobalt—$Co^{2+}$, copper—$Cu^{2+}$, magnesium—$Mg^{2+}$, manganese—$Mn^{2+}$, nickel—$Ni^{2+}$, and zinc—$Zn^{2+}$) that have been found in soil environments, particularly in industrial and agricultural environments where methyl-parathion is used and where MPH enzymes were originally discovered in soil bacteria[33,39,40]. MPH is natively expressed in the periplasm of bacteria, where metal concentrations largely reflect metals present in the environment[41], and the metalation of MPH is likely to have been affected by environmental metal abundance. Thus, our experiment reflects realistic alternative, secondary environments in which MPH adaptation could have occurred. We characterized the complete adaptive landscape defined by five key historical genetic changes—all 32 combinatorial genotypes that separate the ancestral DHCH created by taking the derived MPH genotype and reversing the five historical mutations—under eight different secondary environments. All 32 genes were transformed and expressed in *E. coli* BL21 (DE3), which were grown in cell media supplemented with only one of eight divalent metals (100 μM—a concentration that was selected because it is either equal to or less than the concentrations of each metal ion that have been found in the environmental soil[33,39,40]), and the MPH activity of cell lysate was measured by mixing with methyl-parathion and monitoring the appearance of the *p*-nitrophenol leaving group. We have previously shown that supplementing media with divalent metals in this way does not affect the growth rate of *E. coli* but does impact the activity levels of MPH variants[36]. Note we expressed MPH in the cytoplasm (i.e., the original signal peptide sequence was replaced by strep-tag sequence) to obtain consistent and sufficient expression in *E. coli*. The metal concentrations are likely to be controlled in the cytoplasm, to a certain degree, by homeostasis mechanisms; however, additional supplemental metal in the LB media and in the lysate buffer is sufficient to alter the metalation state of MPH variants and thus their activity level. Still, it is likely that not all intracellular MPH enzymes are acquiring the supplemented metal in the cell (and in particular, some metal ions such as $Ca^{2+}$ and $Mg^{2+}$ may not associate strongly with the enzyme). It is also likely that the enzymes are adopting a mixture of multiple metal-bound states, including each metal binding site accommodating a different metal[36] that may exhibit different catalytic activities. Indeed, the activity levels in cell lysate largely reflect that of purified enzymes for ancestral DHCH and MPH in all metals, while some deviation is observed for $Ca^{2+}$ and $Mg^{2+}$ (Supplementary Fig. 2). Nonetheless, what is clear, and what is most important for our study here, is that these metal environments significantly impact the catalytic activity

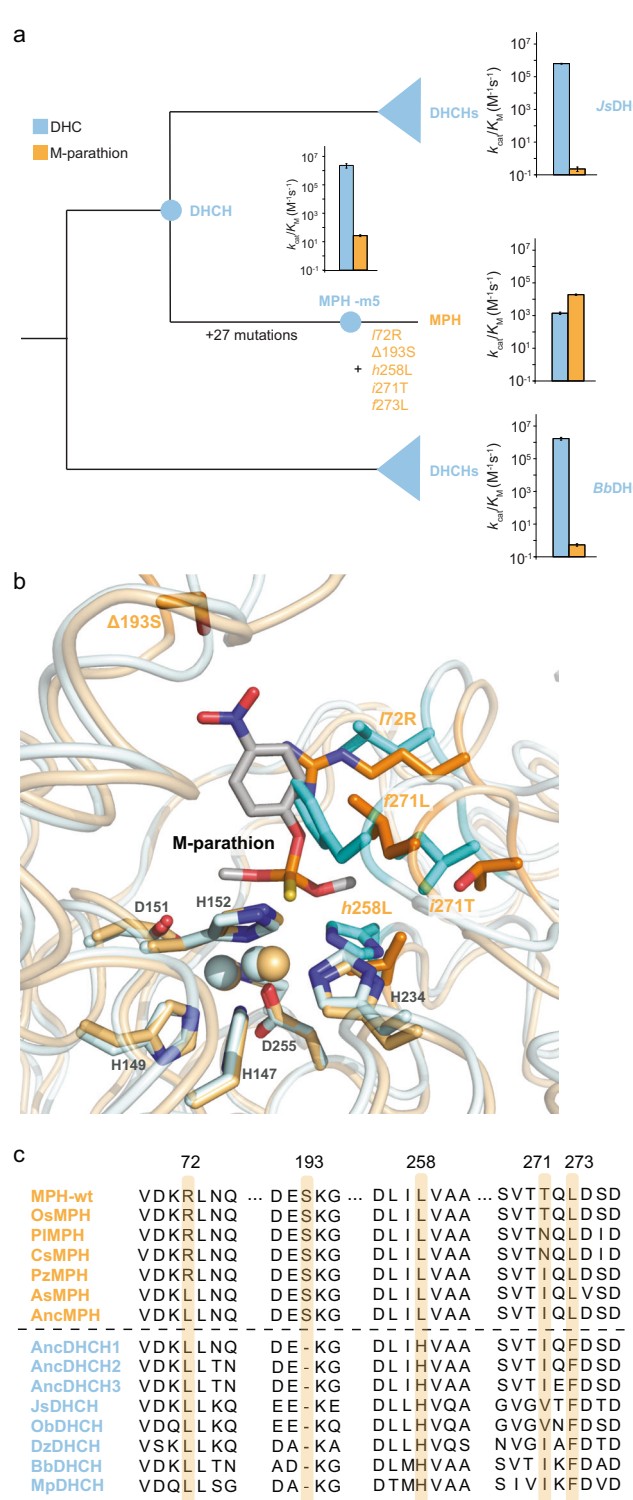

**Fig. 1 Functional evolutionary history of MPH gene family. a** Phylogenetic reconstruction of MPH family and its DHCH relatives. The catalytic activities (kcat/KM) of the enzymes for dihydrocoumarin and methylparathion are displayed in bar graphs (error bars show standard deviation). The five key mutations between the ancestral DHCH enzyme and MPH are labeled in orange on the branch between MPH-m5 and the derived MPH. The schematic phylogenetic was constructed using previously published phylogenetic reconstruction[11]. **b** Overlay of the cartoon representations of the crystal structures of DHCH (cyan, PDB ID: 6c2c) and MPH (orange, PDB ID: 1p9e). The five key mutations are highlighted as sticks and labeled in orange. The two active site metal ions are shown as spheres. Residues involved in coordinating the active site metal ions are highlighted as sticks and labeled in gray. The docking pose of the methyl-parathion substrate is shown as sticks. **c** A cropped multiple sequence alignment of representative sequences of extant MPH, DHCH, and resurrected ancestral enzymes. Residues at the positions where the five active site mutations have occurred between DHCH and MPH are highlighted in orange. **c** is adapted from Yang et al.[11].

more than 100-fold variation in methyl-parathion hydrolysis activity for DHCH and more than tenfold variation for MPH (Fig. 2a). Critically, the change in activity between DHCH and MPH also varies significantly, ranging from ~18-fold improvement (in the $Ni^{2+}$ environment) to 910-fold improvement (in the $Zn^{2+}$ environment), indicating that the effect of the five historical mutations varies significantly depending on the metal environment (Fig. 2b). Second, the effect of even a single mutation in the ancestral genotypic background varies substantially depending on the metal environment (Supplementary Fig. 3). For example, the effect of the $l$72R mutation is positive with seven metals, but negative with $Mn^{2+}$. Similarly, $i$271T had a positive effect in the presence of $Cd^{2+}$, but had a negative effect in all other metal environments. Moreover, whereas $h$258L has a consistently positive effect in all metal environments, the magnitude of its effect varies significantly, ranging from ~18-fold improvement (in the $Cu^{2+}$ environment) up to ~510-fold improvement (in the $Mg^{2+}$ environment) (Supplementary Fig. 3).

The overall topology of the adaptive landscape in each metal environment also differs substantially (Fig. 3). To assess the consequences of this variation for the adaptive process, we applied a simple model of directional Darwinian selection to calculate the most likely trajectory beginning from the ancestral genotype across the adaptive landscape and ending at an "optimal" genotype (i.e., from which all available single mutations would reduce MPH activity—see "Methods" and Fig. 3)[42,43]. Interestingly, the evolution of MPH activity in different metal environments results in trajectories that lead to different optimal genotypes (Fig. 3). For example, trajectories beginning at the ancestral genotype led to the fully derived MPH genotype in only four out of eight secondary environments ($Ca^{2+}$, $Co^{2+}$, $Cu^{2+}$, and $Zn^{2+}$—Fig. 3a, b, e, h). Of the remaining environments tested, three ($Mg^{2+}$, $Mn^{2+}$, and $Ni^{2+}$—Fig. 3c, d, g) maintained the derived MPH as the global optimum across the landscape; however, for each of them the adaptive trajectory that begins from the ancestral DHCH genotype failed to reach it, instead becoming stranded on a local optimum. Finally, in one secondary environment ($Cd^{2+}$) there was a unique global optimum that was not the fully derived MPH genotype (Fig. 3f). Taken together, it is clear that variation in the metal environment can result in varying adaptive landscapes and, as a result, divergent evolutionary outcomes.

**Secondary environmental variation alters mutational effects and key epistatic interactions.** Why do different metal environments produce unique adaptive landscapes and distinct

level of MPH variants and could therefore conceivably impact the topology of the adaptive landscape that results.

**Variation in the adaptive landscape results in divergent adaptive outcomes.** Each metal environment creates a unique adaptive landscape and comparing them highlights several meaningful differences. First, different metal environments result in varying levels of methyl-parathion hydrolysis activity for the fully ancestral (DHCH) and descendant (MPH) enzymes, with

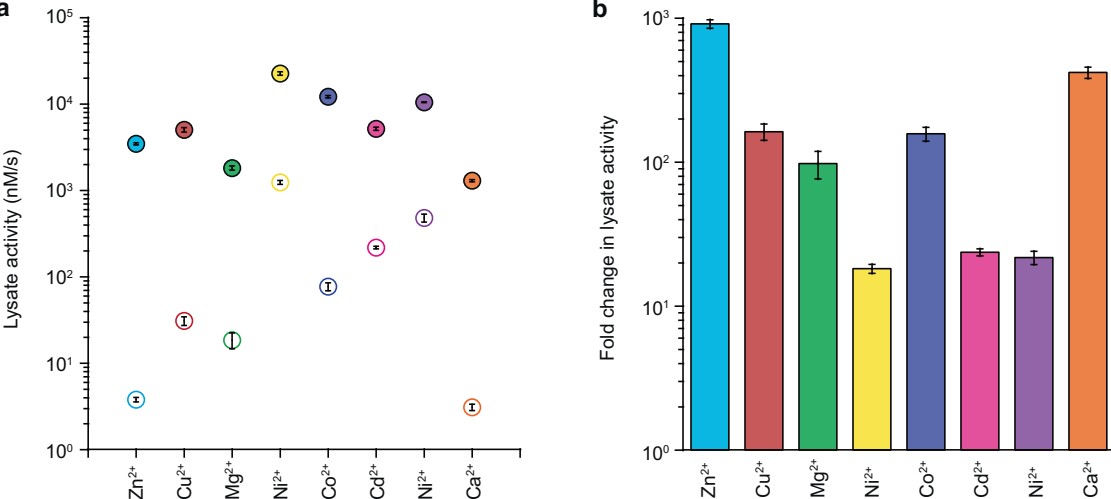

**Fig. 2 Effect of all substitutions when introduced in different metal environments. a** The lysate activity of the ancestral DHCH genotype (open circles) and fully derived MPH genotype (solid circles) in eight different metal environments. Activities shown are the average of three biological replicates, with error bars indicating the standard deviation. **b** The collective effect of all five historical substitutions (fold-change in lysate activity between the ancestral genotype and the derived genotype) in each metal environment. Activities shown are the average of three biological replicates, with error bars indicating the standard deviation. The activities of purified enzymes in the eight different metal environments is presented in Supplementary Fig. 2.

evolutionary outcomes? In particular, what is the molecular basis underlying the unique topology of the $Cd^{2+}$ adaptive landscape? It is expected that evolutionary trajectories can be impacted by two non-additive phenomena: first, genotype-by-environment ($G \times E$) interactions (where the effect of single point mutations changes in different environments)[44] and second, genotype-by-genotype-by-environment ($G \times G \times E$) interactions (which imply that specific epistatic interactions vary depending on the environment)[45]. In order to quantify the impact of metal environment on each historical mutation, we first determined the average effect of each mutation across all possible genotypic backgrounds (see "Methods")[46,47]. As we described previously, the effect of each mutation on MPH activity varies substantially depending on the existence of other mutations, suggesting extensive epistatic interactions among the five mutations[11]. Different secondary environments resulted in qualitatively similar average single-mutational effects, with each mutation usually either increasing ($l72R$, $\Delta193S$, $h258L$, and $f273L$) or decreasing ($i271T$) enzyme activity (Fig. 4). The magnitude of each mutation's effect, however, varied depending on secondary environment. For example, $l72R$ has a highly positive effect in $Zn^{2+}$, $Cu^{2+}$, $Co^{2+}$, and $Ca^{2+}$ environments, but only a marginal effect in $Mg^{2+}$, $Ni^{2+}$, $Cd^{2+}$ environments, and a slightly negative effect in the $Mn^{2+}$ environment, indicating that $G \times E$ interactions at least partially explain variation in the adaptive landscapes. Interestingly, however, the similar average effects of the five individual mutations in all metal environments, including $Cd^{2+}$, suggest that $G \times E$ interactions alone are insufficient to explain the uniqueness of the $Cd^{2+}$ adaptive landscape (Figs. 3 and 4).

Next, we examined the effect of each mutation when introduced into all 16 alternative genetic backgrounds (i.e., its epistatic effects) and performed pairwise linear regression of those effects in different metal environments to assess how well correlated overall epistasis is between environments (equivalent to genotype-by-genotype-by-environment, or $G \times G \times E$, interactions: see "Methods")[48]. Further, we constructed a more complex linear model to fit the adaptive landscape to calculate the degree and contribution of epistasis, including higher-order epistasis, in each metal environment, and determine the impact of the secondary environment on epistasis. The contribution of

epistasis is similar across all metal environments: the first-order effect of mutations explain 68–78% of the overall variation in activity, while between 21 and 31% is attributable to epistasis. However, a model that includes both first- and second-order effects (i.e. average effects and pairwise epistatic interactions) explains between 87 and 97% of the overall variation in activity, while higher-order epistasis (3rd to 5th order) contributes only 0.5–10% collectively (Supplementary Table 2). While the paucity of effect that higher-order interactions appears to have on this function may seem to indicate a relative insignificance in defining evolutionary trajectories, caution should be exercised, as it is in the nature of our nested statistical assessment of more complex models that the highest order effects estimates will be conservative (see "Methods").

When we examine the degree of second-order epistasis, we found significant variation in both the magnitude and the sign (i.e., switching from increasing to decreasing catalytic activity, or vice versa) of specific epistatic interactions across environments (Fig. 5a). For example, the $h258L \times i271T$ interaction is highly synergistic in the $Zn^{2+}$, $Mg^{2+}$, $Cu^{2+}$, and $Ca^{2+}$ environments, but only marginal in the $Mn^{2+}$, $Ni^{2+}$, and $Co^{2+}$ environments, and is highly antagonistic in the $Cd^{2+}$ environment. Similarly, the $l72R \times f273L$ and $i271T \times f273L$ interactions are positive for all metal environments except $Cd^{2+}$ (Fig. 5a). A set of smaller individual effects can explain the difference in other metal environments. For example, smaller average effects of $l72R$, $\Delta193S$, and $f273L$ ($G \times E$) as well as the less synergistic $h258L \times i271T$ and $\Delta193S \times f273L$ interactions contribute to the difference in overall improvement by all five mutations in $Zn^{2+}$ and $Ni^{2+}$ environment (910- vs. 18-fold, Fig. 1c)[24]. These $G \times E$ and $G \times G \times E$ interactions help to explain the unique topology of each metal's adaptive landscape, demonstrating that they can profoundly alter the evolutionary trajectories across it.

As previously described, with the exception of $Cd^{2+}$, all seven other metal environments have the highest activity across this region of sequence space at the fully derived MPH genotype. However, the topology of each landscape is still unique, as each metal environment results in a distinct evolutionary trajectory beginning from the fully ancestral genotype (Fig. 6); while some can clearly evolve to the derived MPH genotype, others could

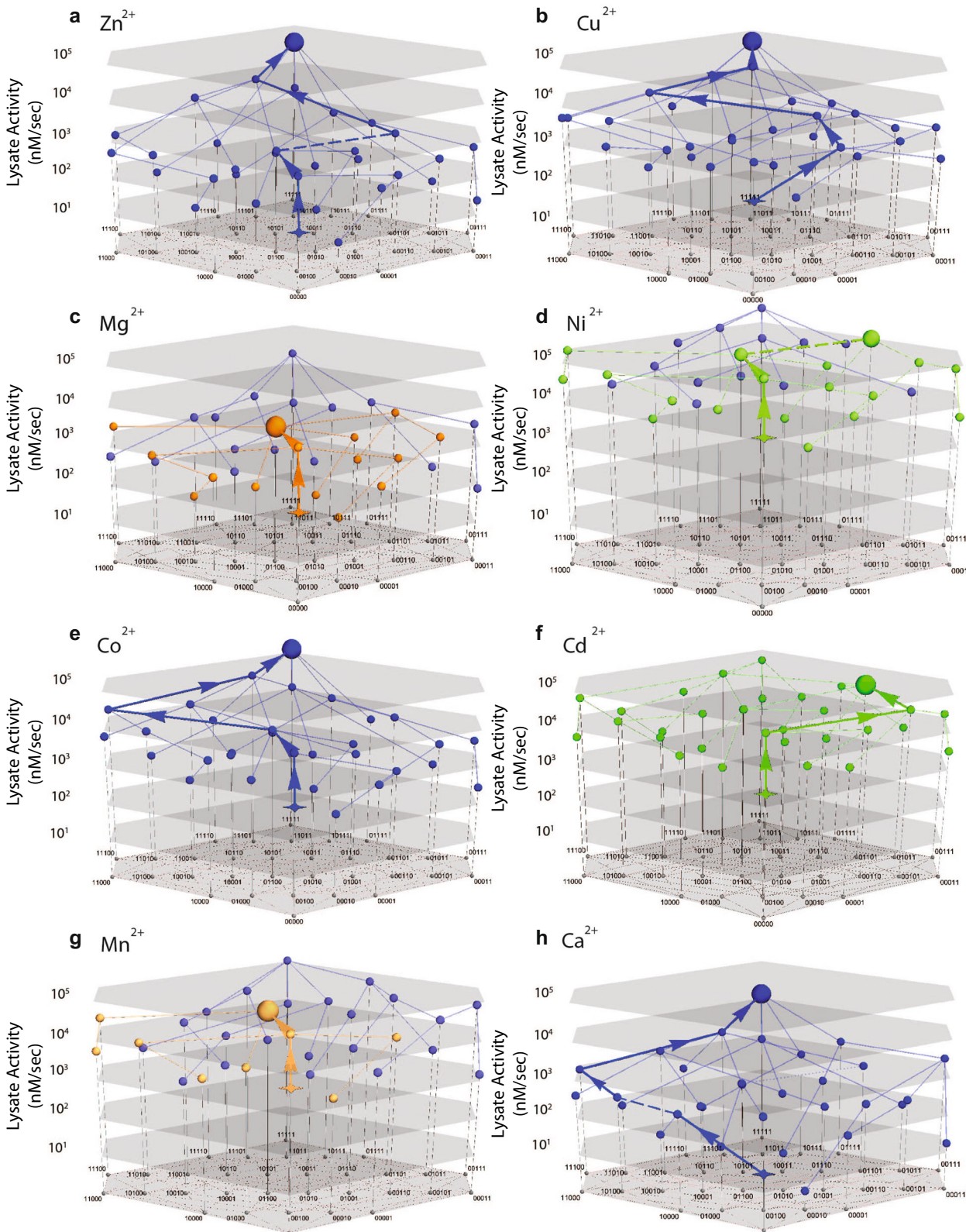

**Fig. 3 Adaptive landscape and likely historical evolutionary trajectories for alternative metal environments. a–h** The adaptive landscape encompassing all 32 genotypes that define this evolutionary transition for all metal environments tested. Local and global optimal genotypes are highlighted with larger nodes while the ancestral genotype (DHCH) is highlighted by a star node. Dashed lines or arrows indicate transitions that are within the margin of error. Blue nodes and lines indicate those that reach the derived genotype (11111); red nodes and lines indicate those that reach the second most common optimal genotype (01100); green nodes and lines indicate those that reach the third most common optimal genotype (01101).

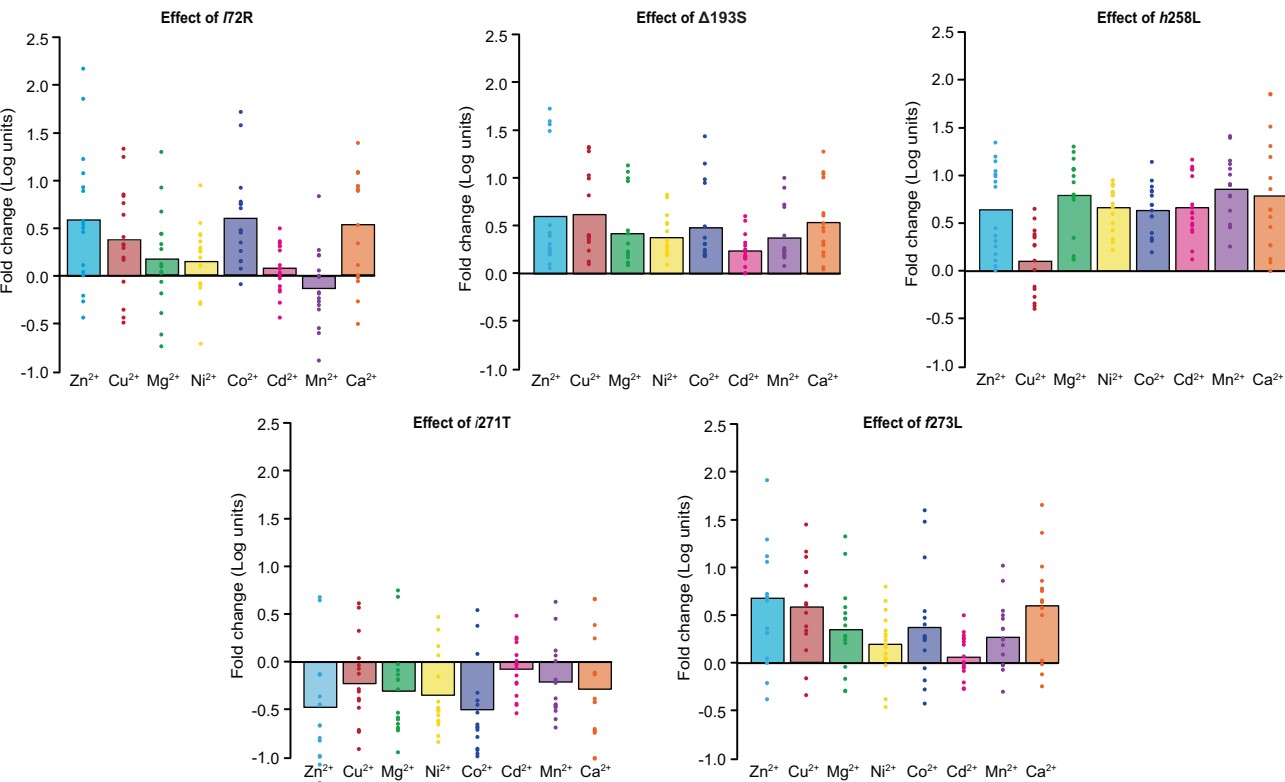

**Fig. 4 The average effect of each of the five historical substitutions when introduced in different metal environments.** Average effect is shown with solid bars while the effect of each mutation introduced in the 16 alternate genetic backgrounds is shown with dots (each bar therefore representing the corresponding average across all 16 dots for each environment).

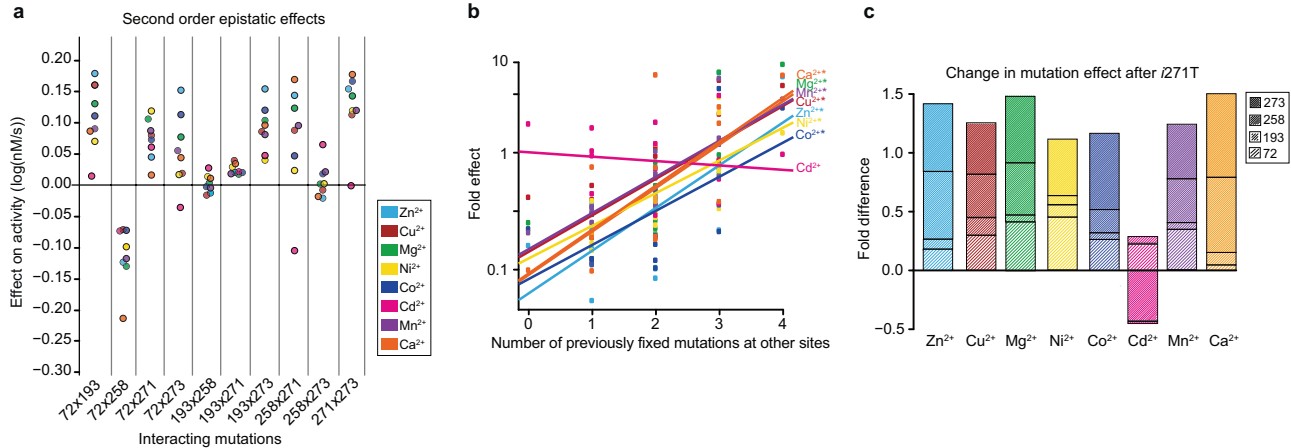

**Fig. 5 Changes in epistatic interactions. a** The pairwise epistatic interaction effects for each metal environment. The two interacting residues are denoted on the x-axis by their positions, with an "x" in between them (e.g., 72 × 193 denote the pair-wise effects of mutations at positions 72 and 193). **b** The relationship between the effect of mutating position 271 and the number of previously fixed substitutions at other sites for each metal. Symbol (*) denotes a statistically significant correlation ($p < 0.05$ after correcting for multiple tests). **c** The impact of previous substitutions at positions 72, 193, 258. and 273 on the effect of the substitution at position 271.

potentially become stranded at a different genotype representing a local maximum instead (Fig. 3c, d, f, g). We analyzed the mutational effects and epistatic interactions that were responsible for these different adaptive landscape topologies. We note several key $G \times G \times E$ interactions that at least partially explain how these landscape differences emerged: $i271T$ is of particular interest, as this mutation reduces activity when introduced into the ancestral genetic

background in all metal environments except $Cd^{2+}$, only becoming positive after several other mutations have first arisen ($i271T$'s effect on activity is positively correlated with the number of mutations that were previously fixed for all environments except $Cd^{2+}$— Fig. 5b). Furthermore, this pattern of $i271T$'s dependence on other mutations is driven by its interactions with $h258L$ and $f273L$ (Fig. 5c). For example, positive epistatic interactions that involve

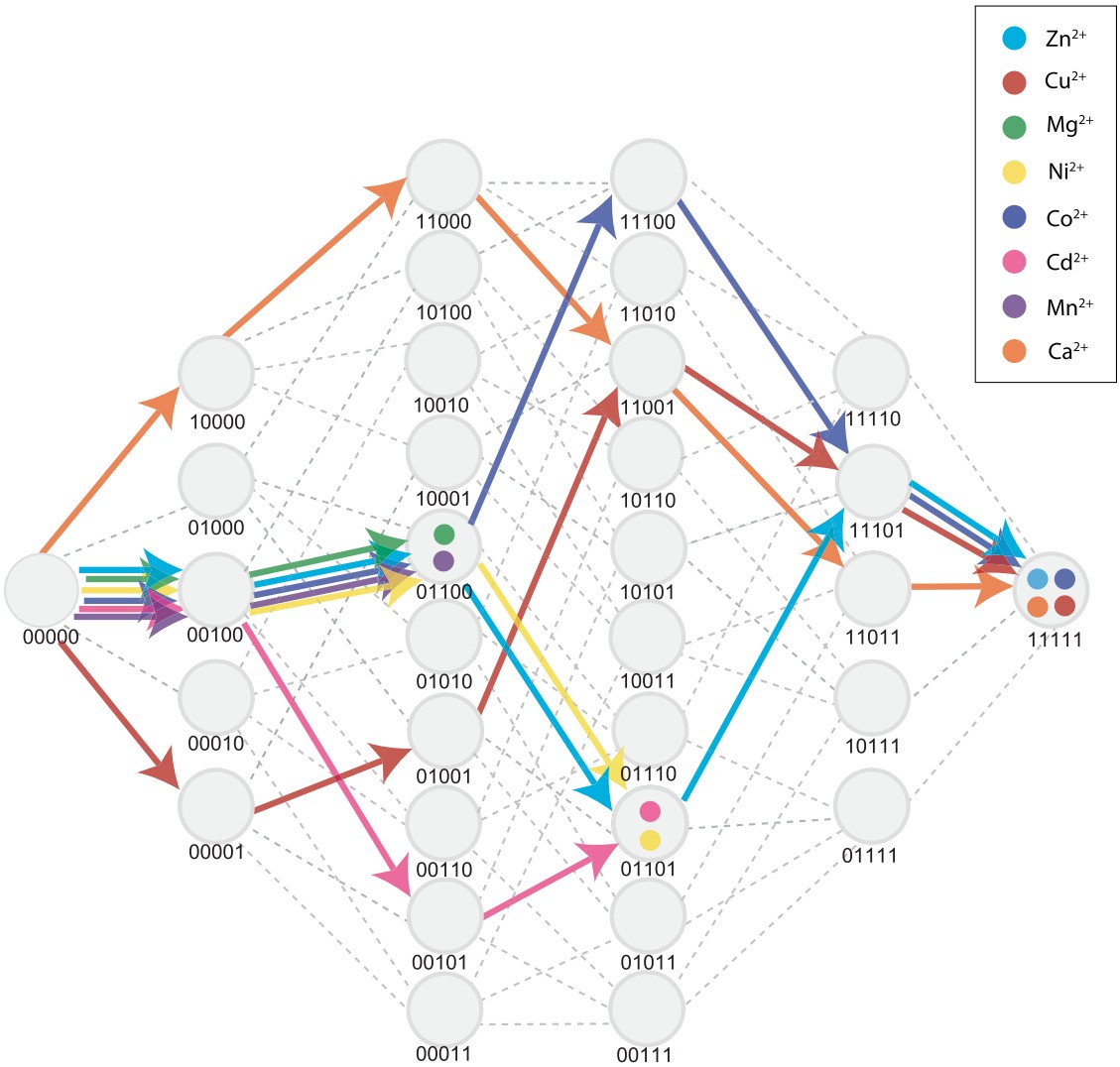

**Fig. 6 Change in functional impact of substitutions along adaptive trajectories.** The impact of substitutions as they are accumulated along the projected adaptive trajectory in each metal environment. Colored dots indicate the corresponding metal environment's "end point" for the projected trajectory beginning at the ancestral DHCH genotype. Genotypes are labeled using a binary system, with "0" indicating the residue being in the ancestral state and "1" indicating the residue being in the derived state and are ordered according to the order of the residues in the enzyme sequence (e.g., the first number indicates the state of position 72, the second the state of position 193, the third the state of position 258 and so on).

$i$271T in the Mn$^{2+}$, Mg$^{2+}$, and Ni$^{2+}$ environments mean that its effect is less negative if introduced after other mutations are already fixed; however, at no point in the projected trajectory are these interactions sufficient to reverse the sign of $i$271T from negative to positive (Fig. 3c, d, g), explaining why adaptive trajectories in those environments fail to reach the fully derived MPH genotype. Similarly, $l$72R × $h$258L exhibits strong antagonistic epistasis, and the fixation of $h$258L leads to $l$72R decreasing enzyme activity in the Mn$^{2+}$ and Mg$^{2+}$ environments, similarly preventing those trajectories from reaching the fully derived MPH genotype (Figs. 5a and 6).

Taken together, the different topology of adaptive landscapes and the existence of some local optima are the result of several different $G \times G \times E$ interactions. In one case, the degree of synergistic epistasis causes an initially negative mutation ($i$271T) to become positive, thus opening a newly accessible mutational pathway. In other cases, antagonistic epistasis causes initially positive mutations ($l$72R and $f$273L) to become negative, thereby restricting potential mutational pathways. This highlights the prominence of epistatic interactions in directing evolutionary trajectories, and demonstrates how even small shifts in the magnitude or sign of those interactions can result in different adaptive outcomes (Figs. 3, 5, and 6).

## Discussion

Overall, this work demonstrates that the secondary, nonselective, environment of metal abundance can significantly alter the topologies of the adaptive landscapes. In particular, our observations reveal several critical details about the effect of metal environmental variation on both adaptive landscapes and the evolutionary trajectories that traverse them. First, alternative metal environments can possess critical differences in both the quantitative measure of an enzyme's function and on the direction and magnitude of mutational effects ($G \times E$ interactions). Second, metal environmental variation can dramatically alter specific epistatic interactions, ($G \times G \times E$ interaction) in some cases causing complete sign reversal between environments. Finally, the consequence of these changes on epistasis and the adaptive landscape lead to changes in the potential evolutionary

outcome[49,50]. In one case, it changes the genotype of the global optimum across this set of evolutionary sequence space, while in others it may instead reach a local optimum as the protein evolves across the adaptive landscape.

What could be the molecular basis for these unique epistatic interactions, adaptive landscapes, and evolutionary outcomes? Our previous work suggested that metal ions in MPH play mostly a functional role rather than a structural role as the apo-enzyme can be generated with chelation treatment in the laboratory[36]. Also, we have shown that the distinct electrostatic properties of the metal ions, rather than any radical change in the active site, caused different activity profiles of the fully derived MPH enzyme by subtly altering substrate and transition state geometries[37], which is also consistent with findings from other enzymes[51]. Moreover, we have also previously shown that these five historical mutations increase the methyl-parathion activity by repositioning the substrate through changing of the shape of the active site cavity[11]. In addition, it is worth noting that the five mutations have not been shown to directly interact with the metal ions in the active site, suggesting that these interactions are likely occurring indirectly through other structural elements of the protein. Thus, none of the genotypic and metal environmental changes drastically alter the mechanism of the MPH's catalysis; instead, it is likely that each subtle change of the electrostatic and/ or active site cavity acts in concert to fine-tune the alignment between the substrate and catalytic machinery. Consequently, changes in even small physical positions can impact key epistatic interactions, thereby altering the topology of the adaptive landscape and leading to different adaptive outcomes.

Our analyses have several shortcomings that are worth noting. First, our experimental model of in vitro cell lysate activity assay (we supply metals in the growth media as well as in the lysate and assay buffers) may not perfectly reflect how enzymes acquire different metal ions and function in the bacterial cell in nature. However, the concentration of most divalent metals in the periplasm largely reflects to the environment due to diffusion via nonspecific porin proteins embedded in the outer membrane[41], and thus enzymes expressed in the periplasmic space, as is the native MPH, could incorporate metals that are abundant in the environment. Thus, our experiment may recapitulate a realistic metalation situation, at some degree, via the effect of environmental metal variation on MPH enzymes. Second, the Darwinian model of strong directional selection for a maximized catalytic function is applied only to the set of five mutations we identified as being responsible for MPH adaptation—in reality, the adaptive landscape would almost certainly have included many other potential mutations, and likely a myriad of alternative potential pathways[52,53]. However, our observations suggest if the evolution of these enzymes was repeated with different metal environments and a larger sequence space was explored, it may lead to substantially different genotypic outcomes.

How common are such shifting evolutionary trajectories by secondary environmental factors likely to be in other systems? At this point, we can only speculate, as data on many more systems must first be collected and analyzed in order to definitively resolve this question. In the case of the DHCH-to-MPH evolutionary transition, the secondary environment of metal abundance is directly linked to replacement of the cofactor in the active site[36,37]. Some metalloenzymes are known to be expressed in the periplasmic space and anchored to the outer membrane, and many have been shown to bind promiscuously to different metal ions that alter their activity profiles[54]. In addition, many enzymes utilize other cofactors and bind to different types of cofactors[55–57]. Moreover, other environmental factors such as temperature, pH, redox potential, salinity, and expression of other proteins such as chaperones, can impact enzyme function and expression, and thus the effects of specific mutations on their function[52,58–62]. Therefore, the secondary environment could easily play a similarly significant role in many other cases of molecular adaptation. If this is the case, it would suggest that rugged and highly environment-dependent adaptive landscapes are the norm and not the exception, likely making evolution even more heavily contingent on minor variation both in environment and in starting genotype than has been previously appreciated[17,18,46,63]. We propose that further studies examining these phenomena should emulate the approach we have used here by examining genetic and environmental effects in concert in order to assess the different shapes that an adaptive landscape may take. Combining the careful construction of statistical linear models and detailed evolutionary pathway analyses under reasonable models of evolution can allow us to more clearly assess the impact that $G \times E$ and $G \times G \times E$ interactions have on evolving proteins. By undertaking this task, we can characterize not only the adaptive landscapes defined by key genetic changes, but we can assess their sensitivity to secondary environmental variation, thereby etching out the sensitivity to alternation of evolutionary outcomes[31].

When there is significant secondary environmental variation and prominent mutational epistasis, evolutionary trajectories can shift, becoming contingent on the conditions in which evolution occurs. Thus, it is critical that we carefully consider the secondary environment as well as the genotypic background in our efforts to predict, design, and understand the evolution of new biological molecules. Adaptation reflects the conditions in which it occurs: its outcome depends both on where it begins and on the landscape across where it travels.

## Methods

**Enzyme cloning and kinetic measurements**. Enzyme genotypes were mutated and cloned into a pET27(b) vector (Novagen) containing a N-terminal Strep-tag II sequence (MASWSHPQFEKGAG) using the *Nco* I and *Hind* III restriction enzymes (Thermo Scientific), as described previously[11]. To test the lysate activities (L.A.s) of variants, *E. coli* BL21 (DE3) transformed with plasmids for each of the 32 MPH variants were grown in triplicates in a 96-deep well plate containing 200 µL of LB media supplemented with 50 µg/mL kanamycin at 30 °C, 900 × rpm overnight. On the following day, a second 96-deep well plate containing 400 µL of LB media supplemented, 50 µg/mL kanamycin, and 100 µM of one of the eight metal ions were inoculated with 20 µL of the aforementioned overnight culture and incubated at 30 °C, 900 × rpm for 3 h. Protein expression was induced by adding IPTG to a final concentration of 1 mM and further incubation at 30°C for 3 h. Cells were harvested by centrifugation at 3320 × g for 10 min and pellets were frozen −80 °C for at least 30 min. To lyse the cells, the cell pellets were resuspended in 200 µL of lysis buffer consisting of 50 mM Tris-HCl pH 7.5, 100 mM NaCl, 200 µM of the same metal ion that was supplied in the LB, 0.1% Triton X100, 100 µg/mL lysozyme, and 1 U/mL of benzonase, and incubated at room temperature with shaking at 1200 × rpm for 1 h. The cell lysates were clarified by centrifugation at 3320 × g for 20 min at 4 °C. To assay enzymatic activity, 20 µL of the clarified lysate was mixed with 80 µL methyl-parathion solution at a final substrate concentration of 400 µM in 50 mM Tris-HCl pH 7.5, 100 mM NaCl, 0.02% Triton-X100 and 200 µM of the same metal that was supplied in the LB and lysis buffer, and the reaction was monitored following the release of *p*-nitrophenol at 405 nm with an extinction coefficient of $18,300 \, M^{-1} \, cm^{-1}$. The L.A. is given as the rate of substrate hydrolysis in nM/s, which is calculated from the molar extinction coefficient of the *p*-nitrophenol leaving group ($18,300 \, M^{-1} \, cm^{-1}$) and normalized to the OD of the cell cultures.

**Enzyme purification and kinetic measurements**. The plasmids containing strep-tagged MPH and MPH-m5 were transformed into *E. coli* BL21 (DE3) and grown in LB with 50 µg/mL kanamycin overnight. The following day, 600 µL of the overnight cultures were used to inoculate 30 mL of 2x YT media with 50 µg/mL kanamycin and 100 µM of one of the eight metals, and the cultures were grown at 30 °C, 280 × rpm for ~3 h. The cultures were subsequently cooled to 16 °C for 30 min, and 0.2 mM of IPTG was added to induce protein expression, and the cultures incubated at 16 °C overnight. Cells were harvested by spinning at 4 °C, 3220 × g for 10 min, and the supernatant removed. For lysis, the cell pellets were frozen at −80 °C overnight, and then resuspended in a mixture of B-PER Protein Extraction Reagent (Thermo Scientific) and 50 mM Tris-HCl buffer, pH

7.5 containing 200 μM of the same metal that was supplied in the 2x YT media, 100 μg/mL lysozyme, and 0.5⁻ U benzonase, and incubated on ice for 1 h. Cell debris was removed by centrifugation at 16,000 × g for 30 min. The clarified lysate was loaded into columns containing about 0.5 mL of Strep-Tactin®XT 4Flow resin (IBA Lifesciences). The columns were washed once with Buffer A (50 mM Tris-HCl, pH 7.5 containing 100 mM NaCl and 200 μM of metal), once with Buffer B (50 mM Tris-HCl, pH 7.5 containing 300 mM NaCl and 200 μM of metal), and a final time with Buffer A. Strep-tagged proteins were eluted with Buffer A containing 50 mM biotin (Sigma-Aldrich), and desalted and concentrated using Microsep Advance Centrifugal Device, 10K Omega (Pall Life Sciences). To assay enzymatic activity, 10 μL of purified enzyme was mixed with 90 μL methyl-parathion solution at a final substrate concentration of 450 μM in 50 mM Tris-HCl pH 7.5, 100 mM NaCl, 0.02% Triton-X100 and 200 μM of the same metal that was supplied in the 2x YT media and lysis buffer, and the reaction was monitored following the release of p-nitrophenol at 405 nm with an extinction coefficient of 18,300 $M^{-1}\,cm^{-1}$.

**Linear modeling of genetic and environmental effects**
*Definition of genetic and environmental encoding system.* To quantify the genetic and environmental determinants of enzyme activity, we used an approach similar to that previously developed[46,47]. We constructed regression models that explain L. A. as a function of the genetic states at the five variable amino acid residues in the protein.

The genetic variation in the protein was defined in the linear models using one-dimensional variables for the mutations; residues 72, 193, 258, 271, and 273 are described by single-dimensional vectors a, b, c, d, and e, respectively, with the ancestral state defined as −1 and the derived state defined as +1 These variables make the y-intercept of the linear model equal to the mean activity across all experimental measurements[47]; therefore, all genetic effects are expressed relative to the mean (Supplementary Table 1).

*First-order linear models.* We constructed our first-order model by regressing the L. A. of each genotype on dependent variables that reflect the individual first-order identities at each genetic position. For example, the linear model for position 72 is expressed as:

$$(L.A.) = a(u_1) + u_0$$

where $a$ is the effect coefficient of moving +1 in that dimension, $u_1$ is the coordinate representing the genotype (i.e., −1 for ancestral leucine, +1 for derived arginine), and $u_0$ is the y-intercept for the model (equal to the mean across the data). The linear coefficients for each model were computed using ordinary least squares regression with the open-source statistical package R (http://www.r-project.org/). The coefficient $a$ indicates the deviation of the derived genetic state from the mean, while −$a$ gives the deviation of the ancestral genetic state from the mean.

To determine how well all five first-order effects of mutations in the protein predict variation in L.A., we constructed the following linear model that included all first-order protein coefficients

$$(L.A.) = a(u_1) + b(u_2) + c(u_3) + d(u_4) + e(u_5) + u_0$$

where $u_2, u_3, u_4,$ and $u_5$ are the coordinates representing the genotype for positions 193, 258, 271, and 273, respectively. We then computed the $R^2$ for this first-order model.

The first-order models for the effect of each environmental factor (i.e., which metal ion was present in the lysate) were modeled using expanded variable space applied along the lines described previously. For this, each metal variable is assigned a unique set of coordinates in seven-dimensional space according to the relevant Hadamard matrix, and those variables were then used to perform a minimal-variable linear regression that is similarly centered to the mean across all the data

$$(L.A.) = f(u_6) + g(u_7) + h(u_8) + i(u_9) + j(u_{10}) + k(u_{11}) + l(u_{12}) + u_0$$

where $u_6, u_7, u_8, u_9, u_{10}, u_{11},$ and $u_{12}$ are the coordinates representing the metal contained in the lysate (full datasets and computational scripts available on Github: DOI: 10.5281/zenodo.4552583). The magnitude of the effect of each metal on L.A. was determined by computing the sum of the modeled coefficients for its defined coordinates.

*Linear models with second-order genetic epistasis and G × E interactions.* To identify cases of second-order epistatic interactions and genotype-by-environment (G × E) interactions, we individually introduced every possible interaction term for every two-way combination of genotypes at the variable sites in the protein or the metal environment. These interaction variables were constructed as previously described[46]. Each interaction is described by a new linear vector, the value for which is determined by taking the outer product between the two first-order linear vectors. For example, the interaction between site 72 and 258 of the protein will be equal to (a) ? (c) = (ac).

Where $u_{13}$ is equal to $u_1u_3$, etc. The second-order interaction effects are equal to the deviation from the additive effect modeled by each genetic state individually across other genetic backgrounds, and is defined herein as the "marginal" effect (i.e., added on to the "average" effects computed in the first-order model).

Interactions between each mutation and the metal environment were modeled analogously. For example, the interaction between site 72 in the protein and the metal environment is constructed by: $(u_1)$ ? $(u_6, u_7, u_8, u_9, u_{10}, u_{11}, u_{12})$ = $(u_1u_6, u_1u_7, u_1u_8, u_1u_9, u_1u_{10}, u_1u_{11}, u_1u_{12})$.

One advantage of this method of encoding the genetic data is that the first-order model is nested within the second-order model. This allowed us to assess whether addition of the second-order model terms significantly improved the fit by comparing the improvement in the adjusted $R^2$ as well as the improvement in the likelihood ratio test relative to the simpler first-order model. The effect of each second-order interaction (i.e., the epistasis and/or G × E interactions that should be added to the sum of the additive lower-order effects) can be solved from these coefficients.

**Evolutionary pathway determination.** To model the evolutionary pathway under a model of strong direction selection (i.e., classic Darwinian adaptation), we developed our own in-house script that calculates the most likely evolutionary pathway under a model of Darwinian selection (https://github.com/danderson8/MPH_Epistasis). Briefly, each genotype's triplicate measurements are considered, and for each genotype the difference in the average activity between it and its single-mutational neighbors (including reversals) is considered for the five evolutionarily relevant changes (specified in the main text). Whichever neighboring genotype provides the greatest improvement in MPH activity (as long as the change is not negative) is selected as the most likely evolutionary "step." An assessment of the confidence in each step is then made by comparing the span of replicate measurements for each of the two genotypes: if they overlap then the mutational step is considered "ambiguous," and it is represented in the trajectory diagram as a dotted or dashed line (see Fig. 3). The "new" genotype is then analyzed in the same way. The process is repeated with each subsequent mutational "step" until an optimal genotype is reached, for which all the neighboring genotypes are significantly (i.e., nonoverlapping) lower in their MPH activity measured, which meets the conditions of being either a local or global optimum across the landscape.

**Reporting summary.** Further information on research design is available in the Nature Research Reporting Summary linked to this article.

## Data availability
All data measurements are available on Github (https://doi.org/10.5281/zenodo.4552583). Source data are provided with this paper.

## Code availability
All custom computer analysis scripts are available on Github (https://doi.org/10.5281/zenodo.4552583).

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

## Acknowledgements
This work was supported by the Natural Sciences and Engineering Research Council of Canada (NSERC) Discovery Grant (RGPIN 2017-04909), and Human Frontier Science Program (HFSP), Program Grant (RGP0054/2020). The authors also gratefully acknowledge helpful comments from members of the Tokuriki lab.

## Author contributions
F.B. and N.T. conceived of original experiments. D.W.A. and N.T. conceived of analyses. G.Y. and F.B. performed wet lab experiments. D.W.A. performed computational analyses. D.W.A. and N.T. wrote primary manuscript with contributions from F.B. and G.Y.

## Competing interests
The authors declare no competing interests.
