## [Peer Review File · Nature Communications]

REVIEWER COMMENTS

Reviewer #1 (Remarks to the Author):

This is an elegant and fascinating study of genotype-x-environment interactions and environment-dependence in GxE interactions, which I recommend for publication. I believe the experiments and statistical analyses are appropriate for answering the scientific question, and that the results are noteworthy.

I have a few suggested additions for improved clarity and context.

The biological context for the enzymes of interest should be explained early in the manuscript, perhaps including the word “bacterial” or more specific detail in the abstract, and some more explanation in the introduction.

The finding that higher-level epistasis played little role in the effects seen may deserve a little more comment (was this expected?), and explicit comparison to the existing literature on epistasis, where the prevalence and importance of higher-order epistatic effects has I believe been a topic of dispute.

Further discussion of the implications for adaptive paths more broadly would seem appropriate. If these results were able to be generalized, what would that mean?

Fig. 1A could be made more informative with more detail, and perhaps a sequence alignment. The meaning of JsDHCH and BbDHCH is not clear, or stated – the reader shouldn’t need to refer to the previous work on these enzymes for the biological context.

There are a number of typographical errors and the manuscript will benefit from careful checking for these, and of the wording in general. Here are some minor suggestions which I believe would improve the writing, to incorporate as deemed fit:

Abstract:

“this variation alters the topology”

“vary in terms of both the number and ...”

p. 3:

line 9: “characterizing”

“often tends to” is redundant – “tends to”

p4:

“we investigate to which extent variation in metal ions changes the effect of individual mutations ...”

“outcomes within”

“where metal concentrations largely reflect metals present in ...”

p.5

“note we expressed MPH in the cytoplasm, i.e. ...”

“and in the lysate buffer is sufficient to”

“each metal binding site accommodating a different metal ...”

p.6

“The overall topology of the ... in each metal environment also differs ...”

“the unique topology of the Cd²⁺ adaptive landscape?”

“First, genotype-by-environment (G×E) interactions (where the effect of single point mutations changes in different environments)”

p. 7

“depending on the existence of other mutations”

p. 8:

“across this region of sequence space”

“only becoming positive after several other ...”

p. 10 “in the cytoplasm”

“abandand” = “abundant”

“our experiment recapitulates”

“However, our observations suggest if the evolution of these enzymes was repeated with different metal environments and a larger sequence space was explored, it may lead to substantially different genotypic outcomes.”

“A some fraction” – this fraction could be specified at least approximately – the sentence would benefit from reworking in any case.

Reviewer #2 (Remarks to the Author):

This manuscript builds upon previous work from the group (Yang et al. 2019) that showed higher-order epistasis between five key mutations was responsible for the evolution of MPH from its DHCH ancestor. The present study demonstrates how environmental factors (different metal ions), in combination with higher-order epistasis, can impact the evolutionary trajectory of the MPH enzyme.

Major concerns:

1. It is not novel or unknown that environmental factors influence the topology of adaptive landscapes. However, this manuscript offers a nice model/example to support this concept. While the work presented is interesting, I am concerned with the novelty and conceptual advance of these findings over previous work from the group (Yang et al. 2019). The earlier paper, published in Nature Chemical Biology, was very comprehensive and featured a variety of both experimental and mathematical modeling techniques. In its current form, this manuscript reads a bit like a “plug and chug” offshoot from the earlier Yang et al. 2019 paper. The modeling methods from the previous study are used with just one additional parameter, different metal ions. Although I appreciate that initial experimental data was used to build the new/revised models, the present study would be significantly strengthened and have greater impact if the models and predicted evolutionary trajectories were supported with additional experiments; i.e. if DHCH lineages with select mutations were evolved in the presence of select metal ions, and the resulting pathways were compared to the predicted trajectories (I do acknowledge that this would be an ambitious undertaking). Therefore, I do not think the manuscript is novel and impactful enough for publication in a journal such as Nature Communications.

2. Anderson et al. claim that the experimental design reflects “realistic secondary environments” and “recapitulates a realistic metalation situation,” and this is not necessarily the case. Despite mentioning that pesticides in industry and agriculture likely contributed to the evolution of the present-day MPH, no environmental metal ion concentrations were used in the study. Instead, the same seemingly arbitrary concentration (100 μ M) was used for all metal ions tested.

3. Similarly, for the lysate assays, MPH was expressed in the cytoplasm rather than in the periplasm, its native location. While this was also done in the Yang et al. 2019 study, this design does not reflect realistic cellular conditions, as metal ion concentrations vary between these two cellular locations.

4. It would be interesting and more informative to determine enzyme kinetics for the MPH variants in the presence of the different metal ions. Assaying enzyme activity using cell lysate probably lacks some degree of sensitivity.

5. The results section of the manuscript has extensive overlaps. The last three sections i.e. Variation in the adaptive landscape results in divergent adaptive peaks, Secondary environmental variation alters mutational effects and key epistatic interactions, and Unique GxGxE interactions lead to different evolutionary trajectories are using overlapping results. The result section should be re-written to make it more concise.

Minor concerns:

1. Overall, the manuscript is not well-proofread. Apart from several typos, some sentences are missing multiple words, which often muddles the message the authors are trying to convey.

2. The title does not accurately reflect the work done and/or the conclusions drawn from the study. While the results suggest that environmental factors likely contributed to MPH evolution, the work does not explicitly demonstrate or characterize the actual conditions that resulted in the present-day MPH

enzyme, as the title suggests.

3. Data should not be discussed or explained/analyzed in the figure legends – this should be relegated to the Results and/or Discussion sections (mostly pertains to the sentences beginning with “Note...”, see: Fig. 3, Fig. 4, Fig 5C).

4. Fig. 1A: The legend should have more description. There is no mention/description of the lysate activity data being shown from the cited previous study.

5. Fig. 2A: Consider re-designing Figure 2A; it is difficult to distinguish between the solid vs. faded circles. Consider using solid/filled circles vs. outlined circles or solid outlines vs. dotted outlines, etc.

6. Fig. 2B: Consider labeling each bar with the corresponding metal ion (in addition to the color coding) to make it easier to read.

7. Fig. 2C: The individual substitutions are not labeled anywhere on the graph (???)

8. Figure 3 is very difficult to follow; maybe it could be replaced with something like fig 6 where it is easier to follow the trajectories.

9. Figure 4: The genetic background in which the mutations are being introduced should be mentioned.

Reviewer #3 (Remarks to the Author):

The presence of new substrates often drives the evolution of new functions. At the molecular level, acquisition of mutations facilitates this molecular evolution; however, the mutational route(s) and the potential interaction among mutations (or existing residues) are poorly understood. One approach taken to delineate the complexity of mutational routes and interactions is to measure the functional activity of all possible genotypes so that one can construct a fitness landscape describing the functional activity across mutational space. Fitness landscapes are extremely useful not only for depicting what is evolutionarily possible (and what is not), but also for studying epistatic interactions among genotypes.

In this manuscript, Anderson and colleagues further elaborated on the transformability of fitness landscapes. They studied the evolution of a dihydrocoumarin hydrolase (DHCH) activity into a methylparathion hydrolase (MPH) activity, which had been largely driven by a set of five mutations (Yang et al 2019 Nat Chem Biol 15:1120). Characterizing the activity of all 32 ($= 2^5$) possible genotypes allowed the authors to construct the fitness landscape for the DHCH-to-MPH transition. Remarkably, the topology of this fitness landscape could be altered by merely supplying different divalent ions to the ancestral DHCH enzyme (and thus metalating the latter). Using a combination of protein expression and statistical analyses, the authors argued that such topological variations had been a result of genotype-by-

environment interactions and environment-dependent epistasis.

Overall, the arguments presented throughout the manuscript is sound and convincing, and the interpretation of the results is adequate. The findings from this manuscript offers both fundamental (e.g. how we understand the evolution of catalysis under different environments) and applied (e.g. engineering new catalysis by utilizing different cofactors) implications for future studies. I have enjoyed reading this manuscript; however, I would like to raise several issues that need to be further addressed or discussed to highlight the wider impact of this work:

1. First of all, it is not easy to comment on specific parts of this manuscript, as it lacks line number and page number.

2. While this manuscript is readable, it needs to be proofread to improve its clarity. There are a few awkward sentences and typos with unclear meaning, e.g.:

- “most fit”  “fittest”? [Introduction]

- “We will discuss the effect of non-selective, secondary environments” (sic)  Is this a complete sentence? [Introduction]

- “abandand”  “abundant”? [Discussion]

- “recapitulate”  “recapitulate”? [Discussion]

3. Epistatic interactions not only “can restrict accessible pathways to the fitness peak”, but can also open up alternative pathways that are otherwise unreachable (Domingo et al 2019 Annu Rev Genomics Hum Genet 20:433). Please clarify this in the Introduction section.

4. The resolution of figures (especially Figs 1, 2, 3, and 5) needs to be considerably improved (or the figure size needs to be larger), as it was difficult for me to clearly interpret the plots. Other related issues concerning the figures are:

- Figure 1: The colors used here need to be described clearly in the figure legend. The structural overlap between DHCH and MPH also needs to be labeled clearly, especially differentiating the residues of MPH from DHCH.

- Figure 2: I am not a fan of the representation of Panel A, and it took me a while to understand what the random dots in the plot mean. In addition, I do not understand Panel C at all. If possible, please simplify and/or present the data in another easily understood manner.

- Figure 3: The y-axis (which I assume to be Fold Changes) needs to be labeled. The resolution of all the plots needs to be improved.

- Figure 4: The bimodal distribution of some metal environments is interesting. I suggest presenting the data as violin plots + internal bars to really highlight the spread of the data.

- Figure 5: The x-axis in Panel A needs be labeled and described clearly. For instance, does “72x193” mean the interaction between mutations at 72 and 193?

- Figure 6: This is a great network plot, but the use of similar color shades for Cu²⁺, Cd²⁺, and Ca²⁺ might be problematic for readers with visual problems. Also, I suggest adding all the genotypes underneath the binary numbers to improve the plot’s clarity and to help the readers to understand all

the possible (and impossible) mutational routes.

5. In addition catalytic support, do the metal ions provide structural support to the enzyme? Also, what is the contribution of metal ions to the free energy of the five mutations? Please discuss and add to the Discussion section.

6. In the same vein, the authors have alluded that metalation with different ions could alter the substrate geometries of the enzyme. Is there experimental evidence for this suggestion? Would it be possible to test the mutant enzymes with several substrates (DHC, MP, and others described in Yang et al 2019 Nat Chem Biol 15:1120) to understand more about the tradeoff effects of metalation with different ions?

7. What are the approximately binding affinities of these ions to the enzyme? Are some more loosely bound than others? Will varying binding affinities affect the extent of metalation and subsequently the hydrolase activity?

8. Figure 2C is particularly problematic as there is no description on the x-axis and I still do not understand the plot. For this reason, the paragraph under “Variation in the adaptive landscape results in divergent adaptive outcomes” is quite incomprehensible for me.

Specific responses

Reviewer #1

The biological context for the enzymes of interest should be explained early in the manuscript, perhaps including the word “bacterial” or more specific detail in the abstract, and some more explanation in the introduction.

We have revised the section of Introduction that described MPH (page 4), and provide more detailed description of biological context for MPH. We have also revised Abstract to highlight that MPH has evolved in bacteria.

The finding that higher-level epistasis played little role in the effects seen may deserve a little more comment (was this expected?), and explicit comparison to the existing literature on epistasis, where the prevalence and importance of higher-order epistatic effects has I believe been a topic of dispute.

While it is relatively small fractions, high-order epistasis contributes at some extent as we described in the text. We believe that explicit comparisons with other examples are beyond the focus of this paper. Nonetheless, we have provided an additional sentence in page 8 to further discuss this matter, and to point out that our methods for statistically characterizing higher-order epistasis is expected to produce conservative estimates for those terms.

Further discussion of the implications for adaptive paths more broadly would seem appropriate. If these results were able to be generalized, what would that mean?

We have provided additional discussion in page 11 to elaborate the implications for environment-dependent adaptive landscapes.

Fig. 1A could be made more informative with more detail, and perhaps a sequence alignment. The meaning of JsDHCH and BbDHCH is not clear, or stated – the reader shouldn’t need to refer to the previous work on these enzymes for the biological context.

We agree with the reviewer, and we have modified Figure 1 accordingly.

There are a number of typographical errors and the manuscript will benefit from careful checking for these, and of the wording in general. Here are some minor suggestions which I believe would improve the writing, to incorporate as deemed fit:

We appreciate the comments and pointing out our typographical errors. We have gone over all typo and further proofread the text.

Abstract:

“this variation alters the topology”

“vary in terms of both the number and ...”

p. 3:

line 9: “characterizing”

“often tends to” is redundant – “tends to”

p4:

“we investigate to which extent variation in metal ions changes the effect of individual mutations ...”

“outcomes within”

“where metal concentrations largely reflect metals present in ...”

p.5

“note we expressed MPH in the cytoplasm, i.e. ...”

“and in the lysate buffer is sufficient to”

“each metal binding site accommodating a different metal ...”

p.6

“The overall topology of the ... in each metal environment also differs ...”

“the unique topology of the Cd²⁺ adaptive landscape?”

“First, genotype-by-environment (G×E) interactions (where the effect of single point mutations changes in different environments)”

p. 7

“depending on the existence of other mutations”

p. 8:

“across this region of sequence space”

“only becoming positive after several other ...”

p. 10 “in the cytoplasm”

“abandand” = “abundant”

“our experiment recapitulates”

“However, our observations suggest if the evolution of these enzymes was repeated with different metal environments and a larger sequence space was explored, it may lead to substantially different genotypic outcomes.”

“A some fraction” – this fraction could be specified at least approximately – the sentence would benefit from reworking in any case.

Reviewer #2

1. It is not novel or unknown that environmental factors influence the topology of adaptive landscapes. However, this manuscript offers a nice model/example to support this concept. While the work presented is interesting, I am concerned with the novelty and conceptual advance of these findings over previous work from the group (Yang et al. 2019). The earlier paper, published in Nature Chemical Biology, was very comprehensive and featured a variety of both experimental and mathematical modeling techniques. In its current form, this manuscript reads a bit like a “plug and chug” offshoot from the earlier Yang et al. 2019 paper. The modeling methods from the previous study are used with just one additional parameter, different metal ions.

In our previous work in *Nature Chemical Biology* (Yang et al 2019), we have identified five historical mutations that underlie the functional transition from an ancestral DHCH enzyme to MPH. We have performed diverse biochemical and biophysical experiments and unveiled the molecular mechanisms of the functional transition. The previous work has indeed provided a foundation for this study, but not a just plug and chug offshoot. In this manuscript, we have set another important question in evolutionary biochemistry, i.e., the effect of secondary environment on a protein evolution, using the MPH evolution as a model system. The concept that environmental factors influence the topology of adaptive landscapes may not be novel and have been discussed among the community, but no systematic experimental study, in particular, with statistical analyses like our study has been reported to date. Additionally, our statistical analysis (which include assessment of $G \times G \times E$ interactions) includes novel extensions from that work (as detailed in **Methods**). The aim and novelty of our manuscript is to study the evolutionary dynamics based on systematic experimental data with strong statistical assessment, rather than qualitative and casual observations.

Although I appreciate that initial experimental data was used to build the new/ revised models, the present study would be significantly strengthened and have greater impact if the models and predicted evolutionary trajectories were supported with additional experiments; i.e. if DHCH lineages with select mutations were evolved in the presence of select metal ions, and the resulting pathways were compared to the predicted trajectories (I do acknowledge that this would be an ambitious undertaking).

The aim of our work is to provide quantitative and statistical picture of the effect of secondary metal environment on an enzyme evolution. We agree with this reviewer's point that an ultimate experiment would be to explore alternative evolutionary pathways in different metal environments by performing parallel laboratory evolution experiments, and further perform quantitative and statistical analyses of mutational landscapes. Unfortunately, our current technology does not allow us to readily perform massive parallel laboratory evolution of MPH, as it is extremely laborious (to such a degree as to be impractical here). We have discussed this limitation in the **Discussion** of our original manuscript. Nevertheless, our extensive statistical analyses of the adaptive landscape of the five historical mutations have provided very clear and surprising pictures for how metal environments can alter mutational effects as well as epistasis between mutations. This finding is novel and provide important implications for protein evolution.

2. Anderson et al. claim that the experimental design reflects "realistic secondary environments" and "recapitulates a realistic metalation situation," and this is not necessarily the case. Despite mentioning that pesticides in industry and agriculture likely contributed to the evolution of the present-day MPH, no environmental metal ion concentrations were used in the study. Instead, the same seemingly arbitrary concentration (100 μ M) was used for all metal ions tested.

We did not intend to present that our experimental conditions are the same environments that the MPH likely evolved. As we discussed in page 5 and 10, we have used a very simplified experimental set up (LB media supplemented with additional metal, overexpression of MPH

variants in *E. coli*, lysate activity assay etc.) which enables us to perform rigorous comparisons between different metal environments. We have set our experimental conditions that MPH can expose to variation of metal ions from the environment. As MPH is originally expressed in the periplasm of bacteria, the metalation of MPH is highly dependent on the environmental condition. Additionally, the concentration of metals using in this study (100 μM) reflects the range of concentrations that many metals are observed in contaminated soils. We have added and modified several elements on page 5 of the revised manuscript in order to clarify these points. Moreover, as we have discussed above and described in Introduction, we use the MPH evolution as a model system to explore the effect of metal environment in the adaptive landscape of an enzyme but we do not aim at exploring “how the historical MPH evolution could have been different if different metal ions are contaminated in the soil.” We have clarified this in discussion too (page 10)

3. Similarly, for the lysate assays, MPH was expressed in the cytoplasm rather than in the periplasm, its native location. While this was also done in the Yang et al. 2019 study, this design does not reflect realistic cellular conditions, as metal ion concentrations vary between these two cellular locations.

This issue was previously raised in the Discussion of our original manuscript. Indeed, it is known that the metal concentrations in the periplasm of bacteria reflect largely environmental concentrations while the cytoplasm is highly regulated by metal homeostasis. Our system (supplying metals in the culture media as well as lysate buffer) mimics the environment of the original MPH, where environmental metals largely dictate the kind of metals associated with proteins. We have added some additional clarification of these points on page 5 and page 10.

4. It would be interesting and more informative to determine enzyme kinetics for the MPH variants in the presence of the different metal ions. Assaying enzyme activity using cell lysate probably lacks some degree of sensitivity.

We agree that obtaining enzyme kinetics for MPH variants in all conditions would provide more information in the enzymology standpoint. However, purifying enzymes would also cause a loss of other information such as a complex metalation state that occur during the protein expression and lysate assay may not maintain during the long purification process. Moreover, performing such experiments (purification and characterizations of 32 variants x 8 conditions = 256 conditions) is extremely laborious in our experimental set up. In order to provide some validation using a direct assay, we have conducted purification and characterization of two variants (ancDHCH and MPH) with 8 different metals, and show that the catalytic activity of purified enzymes largely reflect to that of enzymes in the cell lysate besides Ca^{2+} , which may not bind strongly to the enzyme as we discussed in the original manuscript. We have added this data as Supplementary Figure 3 (a new figure in our revised manuscript), and described and discussed the results on page 5.

5. The results section of the manuscript has extensive overlaps. The last three sections i.e. Variation in the adaptive landscape results in divergent adaptive peaks, Secondary

environmental variation alters mutational effects and key epistatic interactions, and Unique GxGxE interactions lead to different evolutionary trajectories are using overlapping results. The result section should be re-written to make it more concise.

We agree that there are some overlaps between those three sections. We have taken the reviewer's suggestion and consolidated them down to two sections, removing some redundant setup features, while retaining those aspects of the results that were distinct (as the different analyses exposed different aspects of the environment-dependent adaptive landscapes). We believe that analyzing multiple aspects and laying out those results carefully are important to communicate to diverse researchers in the scientific community, in particular to those who are not expert in computational evolutionary biology and statistics. We believe the revised manuscript addresses reviewer 2's concern without compromising our purpose in these sections.

Minor concerns:

1. Overall, the manuscript is not well-proofread. Apart from several typos, some sentences are missing multiple words, which often muddles the message the authors are trying to convey.

We have gone through the manuscript again and attempted to resolve proofreading issues.

2. The title does not accurately reflect the work done and/or the conclusions drawn from the study. While the results suggest that environmental factors likely contributed to MPH evolution, the work does not explicitly demonstrate or characterize the actual conditions that resulted in the present-day MPH enzyme, as the title suggests.

We agree that the title may not reflect our model study, we have modified it to a new title that we believe addresses this reviewer's concern:

The adaptive landscape of a metallo-enzyme is shaped by environment-dependant epistasis

We have modified our figures and figure legends to address the subsequent concerns raised by Reviewer 2.

3. Data should not be discussed or explained/analyzed in the figure legends – this should be relegated to the Results and/or Discussion sections (mostly pertains to the sentences beginning with “Note...”, see: Fig. 3, Fig. 4, Fig 5C).

We have revised all figure legends to precisely describe the figure, and not results and/or discussion.

4. Fig. 1A: The legend should have more description. There is no mention/description of the lysate activity data being shown from the cited previous study.

We have updated Fig 1. We have added the multiple sequence alignment of homologous enzymes, and revised the figure legend to clarify all figures.

5. Fig. 2A: Consider re-designing Figure 2A; it is difficult to distinguish between the solid vs. faded circles. Consider using solid/filled circles vs. outlined circles or solid outlines vs. dotted outlines, etc.

We have revised Fig2A, using the solid and open circles.

6. Fig. 2B: Consider labeling each bar with the corresponding metal ion (in addition to the color coding) to make it easier to read.

We have added the label for each bar.

7. Fig. 2C: The individual substitutions are not labeled anywhere on the graph (???).

We have moved Fig 2C into Supplementary Fig 3 with the same style as seen in Fig 2A.

8. Figure 3 is very difficult to follow; maybe it could be replaced with something like fig 6 where it is easier to follow the trajectories.

We would like to keep Figure 3 as we believe that the 3D landscape like Fig3 is very useful to observe the topological differences between different environments. A presentation like Fig 6 is indeed more useful to describe detailed evolutionary pathways.

9. Figure 4: The genetic background in which the mutations are being introduced should be mentioned.

We have revised Figure 4 including the genetic background and labeling.

Reviewer #3

1. First of all, it is not easy to comment on specific parts of this manuscript, as it lacks line number and page number.

We are very sorry for this inconvenience. We have ensured that page and line numbers are present in the revised manuscript.

2. While this manuscript is readable, it needs to be proofread to improve its clarity.

There are a few awkward sentences and typos with unclear meaning, e.g.:

- “most fit”  “fittest”? [Introduction]

- “We will discuss the effect of non-selective, secondary environments” (sic)  Is this a complete sentence? [Introduction]

- “abandand”  “abundant”? [Discussion]

- “recaptiluate”  “recapitulate”? [Discussion]

We appreciate these corrections by this reviewer. We have corrected all typological mistakes and further proofread to correct our manuscript.

3. Epistatic interactions not only “can restrict accessible pathways to the fitness peak”, but can also open up alternative pathways that are otherwise unreachable (Domingo et al 2019 Annu Rev Genomics Hum Genet 20:433). Please clarify this in the Introduction section.

We have revised the sentences in Introduction (page 3), and cited the reference suggested by R3.

4. The resolution of figures (especially Figs 1, 2, 3, and 5) needs to be considerably improved (or the figure size needs to be larger), as it was difficult for me to clearly interpret the plots. Other related issues concerning the figures are:

- Figure 1: The colors used here need to be described clearly in the figure legend. The structural overlap between DHCH and MPH also needs to be labeled clearly, especially differentiating the residues of MPH from DHCH.

- Figure 2: I am not a fan of the representation of Panel A, and it took me a while to understand what the random dots in the plot mean. In addition, I do not understand Panel C at all. If possible, please simplify and/or present the data in another easily understood manner.

- Figure 3: The y-axis (which I assume to be Fold Changes) needs to be labeled. The resolution of all the plots needs to be improved.

- Figure 4: The bimodal distribution of some metal environments is interesting. I suggest presenting the data as violin plots + internal bars to really highlight the spread of the data.

- Figure 5: The x-axis in Panel A needs be labeled and described clearly. For instance, does “72x193” mean the interaction between mutations at 72 and 193?

- Figure 6: This is a great network plot, but the use of similar color shades for Cu²⁺, Cd²⁺, and Ca²⁺ might be problematic for readers with visual problems. Also, I suggest adding all the genotypes underneath the binary numbers to improve the plot’s clarity and to help the readers to understand all the possible (and impossible) mutational routes.

We apologize for any issues encountered with the Figure resolution. On our home machine, all figures are visible at high resolution; however, we have re-imported them and uploaded them separately (rather than including them in a single combined manuscript), in the hope this will resolve that issue. Additional concerns regarding figure elements have been taken into account as we revised all figures and figure legends in the revised manuscript.

5. In addition catalytic support, do the metal ions provide structural support to the enzyme? Also, what is the contribution of metal ions to the free energy of the five mutations? Please discuss and add to the Discussion section.

The metal ions in MPH are exclusively catalytic support as we have been able to remove metal ions using chelator, apo-enzyme is stable, and reintroduction of metal ions can restore the

catalytic activity. We have clarified this in the Discussion section of the revised manuscript (Page 10).

6. In the same vein, the authors have alluded that metalation with different ions could alter the substrate geometries of the enzyme. Is there experimental evidence for this suggestion? Would it be possible to test the mutant enzymes with several substrates (DHC, MP, and others described in Yang et al 2019 Nat Chem Biol 15:1120) to understand more about the tradeoff effects of metalation with different ions?

The molecular mechanism underlying different metalations is a very interesting point to address indeed. In fact, we have performed some preliminary experiments for testing other organophosphates with different metal ions. However, it is very challenging to extract and interpretate the both metal and substrate effects from the massive experimental data. In order to provide further advanced conclusion beyond our discussion, we feel that much more additional experiments (crystal structures with different metals etc) would be necessary to gain insights into the deeper molecular mechanisms of metalation and different mutational effects. We feel that such works would be too much extension, and beyond the scope of this work. Thus, we would aim to engage in this work in a future extension from this project.

7. What are the approximately binding affinities of these ions to the enzyme? Are some more loosely bound than others? Will varying binding affinities affect the extent of metalation and subsequently the hydrolase activity?

We have not measured the binding affinities of each metal ions. Unfortunately, a preliminary trial for ITC experiments has not provided a clean data to quantify the metal affinities. As we discussed above, the mechanistic aspect of the metalation and catalytic activity of MPH variants is not the scope of this work, and we would like to take this suggestion in our future work. Nonetheless, we have discussed some aspect of the metal affinity in page 5.

8. Figure 2C is particularly problematic as there is no description on the x-axis and I still do not understand the plot. For this reason, the paragraph under “Variation in the adaptive landscape results in divergent adaptive outcomes” is quite incomprehensible for me.

We have revised Figure 2 and removed the panel in 2c, instead replotting the data in what we hope is a more clear fashion in Supplemental Figure 3.

REVIEWERS' COMMENTS

Reviewer #1 (Remarks to the Author):

The role of different kinds of epistasis in evolution is one of biology's central questions. Anderson et al. quantify the effects of GxGxE interactions, using gene variants of a metallo-enzyme expressed in *E. coli* under different environmental conditions (media supplemented with different divalent metal ions).

The revised paper is stronger than the initial submission, after having responded to important reviewer points. The explanation of GxGxE interactions, a difficult topic, is I think clear and helpful. I also appreciate the comments on the broader implications of the study regarding the potential general contingency of evolutionary histories and the multiple citations to papers dealing with fundamental issues in this area – the extent of 'contingency' is another central question for evolutionary theory, like that of the role of epistasis, and this study while necessarily only preliminary, sheds real light on both.

Potential further improvements:

The results in Figure 1a are striking when understood, but not immediately clear from the figure. I think that Figure 1a should be labelled more clearly still; in particular, clarify that the colours in the key refer to the substrates. Different colours for the substrates DHC [e.g. green] and MP [e.g. pink] versus the enzymes DHCH [e.g. cyan] and MPH [e.g. orange] would be useful, with more labelling to emphasise that the derived MPH is the intended focal point of the figure.

For a clearer figure, some suggestions: the y axes titles could also perhaps be left off all but one graph. Horizontal rather than vertical bars could also be considered, and it may be helpful to re-order the phylogenetic tree such that the derived MPH and intermediate MPH-m5 are either at the top or bottom of the figure rather than the middle.

Likewise, the meaning of the colours in 1b and 1c could be clarified with labels.

The apparent trade-off between hydrolysis of DHC and MP substrates in the evolution of the derived MPH enzyme (i.e. less effective DHC hydrolysis) observed in Figure 1a may be worth noting, as it shows the "enzyme activity" referred to in the paper has additional unexplored facets. Following on from this, the question of GxGxE interactions with regards to the ancestral substrate could be mentioned as an unanswered question. I take it that the p-nitrophenol leaving group of hydrolysed methyl-parathion makes it an attractive study substrate as compared to DHC; this key aspect of the experimental design could be clarified.

Minor points:

p. 3, line 23: "epistasis between mutations" doesn't seem the right term for GxGxE – I would interpret this phrasing as meaning GxG.

p. 4, line 28 – citations here have a formatting glitch, and I presume are meant to be in the first

sentence.

p. 4, line 29 “do not impose”

Zachary Ardern

Reviewer #3 (Remarks to the Author):

The authors have addressed my previous comments adequately and incorporated the necessary changes in this revised manuscript. In addition, the submitted codes are satisfactory (as I have managed to reproduce the calculation outputs using the authors' scripts).